# COVID-19-Related Racial Discrimination during Lockdown and Its Impact on Asian American Women

**DOI:** 10.3390/ijerph20166546

**Published:** 2023-08-08

**Authors:** Mina Lee, In Young Park, Michael Park, Phuong Khanh Tran, Yvette C. Cozier, Hyeouk Chris Hahm

**Affiliations:** 1Department of Social Work, College of Community and Public Affairs, Binghamton University, Binghamton, NY 13902, USA; mlee48@binghamton.edu; 2School of Social Work, Boston College, Boston, MA 02467, USA; inyoung.park@bc.edu; 3School of Social Work, Rutgers University, New Brunswick, NJ 08901, USA; p.michael@rutgers.edu; 4Department of Electrical and Computer Engineering, College of Engineering, Boston University, Boston, MA 02215, USA; kontact@bu.edu; 5School of Public Health, Boston University, Boston, MA 02215, USA; yvettec@bu.edu; 6School of Social Work, Boston University, Boston, MA 02118, USA

**Keywords:** Asian American women, racial identity, COVID-19 pandemic, racial discrimination

## Abstract

During the lockdown period of the COVID-19 pandemic, Asian American (AA) women have experienced a surge in anti-Asian hate crimes and racial discrimination, and a majority of studies have quantitatively shown the negative impact of these incidents on Asian Americans’ well-being. Our research expands on the existing literature by qualitatively investigating types of COVID-19-related racial discrimination during lockdown and its impacts on changes in emotions, behaviors, well-being, and racial identity development among AA women. This study covered two timepoints (December 2019 to May 2020) and the data were collected using an open-ended survey with 40 AA women. Thematic analysis identified core themes related to types of racial discrimination, emotional and behavioral changes, and racial identity status that emerged due to COVID-19-related racial discrimination experiences. The findings shed light on the long-lasting impacts of racial discrimination on AA women’s overall well-being and dynamic development of racial identity. Altogether, our findings underscore the need for systematic forms of advocacy to combat anti-Asian racism and call for solidarity for AA women’s well-being.

## 1. Introduction

### 1.1. COVID-19-Related Racial Discrimination

The COVID-19 pandemic has caused surges in the number of hate crimes and the amount of racism against Asians in the United States, predominantly in 2020 and 2021. The number of reported anti-Asian hate crimes in the US sharply increased by almost 80% nationwide [1] and by more than two-fold in larger cities [2]. In particular, Asian American (AA) women have been the prime targets of hate crimes and racial discrimination during this period, reporting a 2.3 times higher rate of hate crimes than AA men [2]. The Atlanta spa shootings on 16 March 2021, in which six of the eight victims were AA women, is one example of the seriousness of anti-Asian hate crimes [3]. It is likely that racial discrimination against AAs became more violent during COVID-19, and those who are physically more vulnerable such as women and the elderly were more susceptible to racial discrimination. Research consistently shows that racist incidents generally lead to deleterious effects on mental and physical health and racial identity among all racial minorities [4,5]. Furthermore, AA women are more likely to be directly impacted by racial discrimination than AA men [6]. With the surge in COVID-19-related racial discrimination targeting AA women, investigations that explore the nature and impact of COVID-19-related anti-Asian discrimination are necessary.

In US history, AAs have faced various types of systematic and overt racial discrimination, ranging from the Chinese Exclusion Act of 1882 and the internment of Japanese Americans during World War II [7] to microaggressions in everyday life [8]. In recent decades, pre-pandemic anti-AA racial discrimination had evolved into more subtle, ambiguous, and covert forms (e.g., microaggressions) and was ubiquitous [9]. However, since the start of the COVID-19 outbreak, and political comments from the US executive branch, racial discrimination has shifted from microaggressions to direct, overt, and violent forms.

Studies have reported different types of racial discrimination experienced by AA women in different contexts. Specifically, one study demonstrated that AA experienced COVID-19-related racial discrimination at the societal, interpersonal, and intrapersonal levels [10]. Another study showed that Chinese American parents experienced six different types of racial discrimination during the pandemic: online direct (e.g., directed at the individual), online vicarious (e.g., directed at others), in-person direct, in-person vicarious, health Sinophobia, and media Sinophobia [11]. Further, a study found that after the COVID-19 outbreak, hate crime-related topics increased on an online platform where physical difference had been the prevalent topic prior to the pandemic [12]. While these are noteworthy findings, these studies have predominantly utilized quantitative methods, making it difficult to capture the voices of AAs more directly, including the details of their experiences of COVID-19-related racial discrimination. Thus, qualitative studies are needed to better understand the types of racial discrimination that may be unique to AAs.

### 1.2. COVID-19-Related Racial Discrimination and Its Impact on AA Women

Studies have shown that perceived racial discrimination has negative impacts on psychological well-being and racial identity among all racial minorities [12,13]. Similarly, COVID-19-related racial discrimination has contributed to mental distress among AAs, including depressive symptoms [10,14,15,16,17], anxiety [10,18], and psychological distress [19,20]. A multiracial qualitative study reported that COVID-19-related anti-Asian discrimination caused fear, anxiety/distress, hopelessness/depression, and avoidance in all participants, regardless of their race [10]. To our knowledge, no academic study to date has qualitatively investigated the impact of COVID-19-related racial discrimination on emotion, behavior, and general well-being exclusively among AA women.

Racial identity is defined as a subjective connection or affinity within a socially designated racial group [21]. Specifically, AA racial identity is externally defined and is based on the sociopolitical construction of the Asian race that includes more than 30 ethnicities [20,21]. AAs develop their racial identity as members of pan-ethnic AA communities as well as individual ethnic groups. Despite the agreement that racial identity is affected by environmental factors, there are competing theories on the direction in which racial discrimination impacts racial identity development among AAs. For example, the AA Racial Identity Development model (AARID model) suggests that racial discrimination may weaken racial identity among AAs because it has led them to uncritically endorse dominant values, beliefs, and standards, including negative attitudes and stereotypes, towards their ingroups [22,23]. In contrast, the Rejection–Identification model (RIM) suggests that racial discrimination strengthens racial identity among racial minorities as it increases the salience of race among social identities [24]. There is no strong empirical support for RIM among AA women. Considering the inconsistency in the literature about the direction of the impact of racial discrimination on racial identity, it is particularly important to investigate how COVID-19-related racial discrimination shaped racial identity among AA women. To our knowledge, this is the first study to qualitatively examine the impact of COVID-19-related racial discrimination on racial identity among AA women in the US context [15].

### 1.3. Current Study

Despite the strength and breadth of research examining the impact of COVID-19-related racial discrimination on health and mental health, the surge in prevalence and severity of anti-Asian discrimination during the pandemic warrants further examination. This study aims to explore the types of anti-Asian racial discrimination experienced during the COVID-19 lockdown and its impact on emotional and behavioral health, well-being, and racial identity in a sample of AA women. Its qualitative approach allows for flexible participant engagement and the potential to deepen our understanding of the impacts of racial discrimination in both anticipated and unanticipated areas.

## 2. Materials and Methods

### 2.1. Study Design

This study utilized Waves 1 and 2 of the Epidemiology Asian Women’s Action for Resilience and Empowerment (Epi AWARE) study, a longitudinal study examining the health of young Asian women. Wave 1 of Epi AWARE was collected between 6 December 2019 and 12 March 2020 and corresponds to the pre-pandemic period in the US. Wave 2 was collected between 24 April and 15 May 2020, a period characterized by stay-at-home orders in effect in many states and rising politically based anti-Asian sentiment, and hate crimes [1,2].

The Epi AWARE cohort consisted of two groups of women: participants in the AWARE study and women recruited from the Boston area. **(1) The AWARE study.** The Asian Women’s Action for Resilience and Empowerment (AWARE) study [10]) was an NIH-funded randomized intervention study regarding Asian women’s mental and sexual health (PI: Hahm) conducted from 2018 to 2019. AWARE study participants were self-identified women aged 18–35 years, unmarried, 1.5- or 2nd-generation immigrants from China, Vietnam, and Korea, and residents within the Greater Boston (MA, USA) area. An invitational letter containing a link to the Epi AWARE Wave 1 online questionnaire was emailed to all AWARE participants with an email address on file as of November 2019. **(2) Local recruitment.** AA women were recruited to the study through printed flyers placed on local (Boston, MA, USA) college campuses and community centers, advertising on social media email listservs, and word of mouth. Interested women contacted the study and were subsequently emailed a letter containing a link to the Epi AWARE Wave 1 online questionnaire. In order to engage a more diverse group of AA women, study eligibility criteria were expanded to include self-identified AA women aged 18 and older of any generational status.

Initially, Epi AWARE investigators planned to collect questionnaire data in two waves: at baseline (Wave 1, beginning December 2019) and follow-up at 6 months (originally designated as Wave 2, beginning June 2020). However, during the first wave of data collection, the COVID-19 outbreak occurred, followed by a lockdown and a surge in anti-Asian hate crimes. In response to these environmental changes, the research team sought to understand the impact of this historical moment within our cohort by asking participants open-ended questions related to their contemporary experiences (newly designated as Wave 2). After the open-ended questions were collected, the team resumed its original plan of collecting follow-up data (now Wave 3) in September 2020. This research project was approved by the Institutional Review Board of the affiliated University Medical Center.

### 2.2. Epi AWARE Questionnaires

The Wave 1 questionnaire collected information on demographic (e.g., age, ethnicity, marital status), medical, and lifestyle factors, and race consciousness and experiences of discrimination.

The Wave 2 questionnaire posed open-ended questions to participants about their personal experiences, which resulted in free-text data. The Qualtrics survey asked participants about COVID-19-related racial discrimination experiences and their impact on emotions, behavior, and racial identity. Specifically, the questions included the following: (1) Have you experienced racial microaggression, physical assault, or discrimination related to COVID-19, (2) Has someone you are close to experienced racial microaggression or discrimination related to COVID-19, (3) How did COVID-19 experiences affect the way you perceive your own racial identity as Asian, (4) Overall, how have these experiences changed your feelings, behaviors, and general well-being before and after COVID-19, and (5) In your opinion, what are the necessary steps or methods to mitigate the growing racism? In this study, the data generated from questions 1 through 4 were explored.

### 2.3. Analysis

The demographic information of participants, including age, ethnicity, nativity, educational attainment, marital status, and household income were descriptively analyzed. The frequency and proportion of direct and vicarious racial discrimination among respondents were examined. The proportion was calculated in relation to the total sample size and not the total number of respondents to the questions; therefore, the frequency and proportion may not add up to the total sample size and 100%. 

The data were divided into four different datasets according to the question to which responses were provided and then analyzed separately. Free-text responses in Wave 2 data were analyzed using thematic analysis consistent with studies analyzing free-text survey responses [25,26]. Thematic analysis is a high-level analytic approach that enables researchers to analyze open-ended data and identify overarching patterns that emerge from the data [25]. Two researchers (ML and IYP) familiarized themselves with the data by reading all the responses repeatedly. Each researcher independently came up with initial codes that captured the most prevalent patterns emerging from the data. Through consensus meetings, two researchers (ML and IYP) created an initial codebook that includes the definitions and examples of each code and multiple codes given to individual responses as long as the codes appropriately cover the contents of responses. The codes of two researchers were agreed upon 85% of the time, and inconsistency was resolved through further discussion in which the other authors (MP and HH) gave their third opinions. Codes were modified and finally generated into higher-order concepts (i.e., themes and major themes) based on their common properties through the iterative discussions within the study team.

## 3. Results

Forty respondents provided written answers for the open-ended survey, resulting in a data corpus of 4323 words. The extent of the data ranges from a few words to longer texts with several sentences.

Table 1 presents the demographic characteristics of the study participants. A total of 40 AA women who participated in Wave 1 also completed the Wave 2 survey. The age of participants ranges from 18 to 57 with a mean of 29 years old. Approximately 70% of the respondents were born in the US, and 80% had never been married. In addition, the sample was overwhelmingly educated and affluent, with nearly 80% of participants holding undergraduate or graduate degrees and 40% earning more than $100,000 a year in the previous year.

More than half of the respondents reported that they experienced either direct or vicarious racial discrimination during COVID-19. Fourteen respondents reported that they experienced direct racial discrimination and nineteen respondents reported that someone close to them experienced it. About 25% of our sample experienced both direct and vicarious racial discrimination.

### 3.1. Racial Discrimination and Microaggressions Experienced by AA Women

The themes that emerged in the responses about experiences of racial discrimination and microaggressions are illustrated with quotes in Table 2. The most common type of racial discrimination was a *verbal attack* in a public space. Respondents reported that they were yelled at, and some offenders intentionally approached them to yell at them. In other cases, the offenders were driving by in a vehicle while the respondent was walking on the street. Typical racial slurs and mocking Chinese words were yelled at the respondents, as well as phrases such as “go back to your country”. In addition to these typical expressions of anti-Asian discrimination, the respondents also experienced verbal attacks that were related to COVID-19. Respondents reported the words “coronavirus”, “dirty Asians”, and “China virus” being said, and these words sometimes got yelled directly at them. Respondents also heard comments from strangers that wrongly assumed or accused them of being sick.

Another common type of discrimination or microaggression that AA women experienced was distancing and avoiding, and these experiences are particularly related to the COVID-19 outbreak. Women reported being asked to keep their distance or stay where they were; they were also avoided by strangers in public spaces. One respondent reported being asked to maintain twice the officially recommended six feet of social distancing. Three respondents reported that these situations occurred in grocery stores, which are inevitably necessary places. Two of the three respondents specified that the offenders were white individuals.

Two other types of discrimination and microaggressions the respondents experienced were *staring and being followed*. Five respondents reported that they experienced “lingering stares” and “awkward and dirty looks” for no apparent reason. Two respondents reported that they were followed by others while being verbally attacked.

The last theme that emerged among AA women who experienced discrimination and microaggressions was *online discourse*. The respondents perceived online comments such as the “China virus” as the main source of discrimination and microaggression. A respondent reported that comments that were meant to be “joking” were still hurting her.

### 3.2. Changes in Emotion, Behavior, and Well-Being

Four different themes emerged from the responses related to the self-reported changes in their emotions, behavior, and well-being (Table 3). The four themes are (1) emotional strain, (2) social anxiety and limited social engagement, (3) concern for other Asians, and (4) desire for solidarity.

Experiences of *emotional strain* were the most common change that the respondents reported. The *emotional strain* included feeling hopeless, stressed, worried, angry, frustrated, afraid, paranoid, nervous, and disappointed. Some emotional strain came from direct experiences of discrimination, while some reported concerns for potential victimization.

Some respondents experienced social anxiety, which consequently limited their social engagement either because of direct experiences of racial discrimination or frequent reports of incidents against AAs in general. This theme shows that the impact of discrimination extends beyond emotions, resulting in negative (avoidant) behavioral change. Respondents reported that the current racial atmosphere made them concerned about how other people perceive them and this concern blocked them from actively engaging in daily social interactions and settings. For example, some respondents were scared to interact with people, found it harder to smile or say hello when encountering strangers, and did not leave their houses. Some respondents reported that they only went to Asian-owned businesses or Asian grocery stores to avoid potential victimization. Finally, one respondent reported that she felt she had to compromise her own health by not wearing a mask because she was afraid of being the target of violence, even though she knew that wearing a mask would protect her.

The remaining two themes are more positive than the previous two: *concerns for other Asians* and *desire for solidarity.* Respondents expressed concern for their close family members because of the potential violence that they may experience. For some respondents, this concern extended to AAs in general and those who have been the target of hate crimes. Furthermore, the theme of desire for solidarity shows that some AA women wanted to stand up and speak out against all racism. One respondent specifically expressed that her desire to resist racism goes beyond the AA community and extends to all ethnicities.

### 3.3. Changes in Racial Identity

Four themes emerged regarding self-reported changes in racial identity: (1) race hyperconsciousness, (2) the awakening to social–political positionality, (3) resiliency, and (4) empowerment (Table 4). The largest number of respondents reported that they became hyperconscious about being an AA after the COVID-19 outbreak and related anti-AA racial discrimination. Race consciousness refers to heightened awareness of one’s stigmatized status [27]. Many respondents said that they became hyperaware, paranoid, and cautious about becoming a target of racially oriented violence in public spaces. Others revealed that they were concerned about being a target of racial discrimination during COVID-19 regardless of their prior experiences of race-related discrimination.

Eleven individuals reported that being marginalized and othered made them realize their position in the US racial hierarchy for the first time. Reflecting the triangulated racial positionality of AAs [28], respondents reported that they were perceived as inferior and were othered by non-Asians. For many respondents, COVID-19-related racial discrimination awakened them for the first time to others’ perceptions about AAs. This awareness caused some to express frustration because they and their families were born in the US and “tried to assimilate into culture”. One respondent also reported that it exposes the fragility of the model minority myth.

Despite the surge in COVID-19-related racial discrimination, some reported that their AA identity is *resilient*. Respondents consistently made it clear that their racial identity was not affected by the current surge in racial discrimination against AAs, but two distinctive subconstructs emerged. The first is related to the idea that their AA identity is so strong that it cannot be weakened by external circumstances. The second is related to the idea that racial discrimination against AAs has always been present, and that the surge is not something new, and does not affect their racial identity as AAs. Thus, prior experiences of racial discrimination may have been a protective factor against the surge in racial incidents during the early stages of the pandemic. One respondent revealed having “lived in the Deep South”, where she was discriminated against “by both white and black people daily”.

The final theme that emerged among participants after experiencing COVID-19-related racial discrimination is *empowerment*. By understanding how others see them, their racial identity as AA became salient to some AA women and was empowering. Some respondents expressed the desire to stand up against racial comments and to build connections with other AAs, Asians in Asian countries, and other racial minorities. As they recognized that they had similar difficulties, AA women found it necessary to connect with other Asians who would similarly need support. One respondent who expressed a desire to stand with other AAs who experienced discrimination mentioned that she also wanted to “stand with other folks of color who experience discrimination.” In addition to feeling solidarity, it translated into a proactive fight against racism and the support of Asian communities. One respondent stated that she “should be a voice standing up against the racist comments”.

## 4. Discussion

This study investigated the types of racial discrimination and microaggressions that AA women experienced during the COVID-19 pandemic lockdown and the impact on their well-being and racial identity. The study’s strength of using qualitative methods elicited descriptions of the experiences of AA women in their own words.

During the COVID-19 lockdown, the covert types of racial discrimination against AAs that have always been prevalent continued. Study respondents reported experiencing lingering stares, making them feel uncomfortable and as if they did not belong in those spaces. More importantly, AA women experienced overt types of racial discrimination during the lockdown period that were more aggressive and potentially violent. They were called typical racial slurs and were the targets of contemptuous comments highlighting common foreigner stereotypes and their otherness. Racial slurs were sometimes accompanied by intimidating behaviors such as being followed by a car.

In addition, AA women experienced racism unique to the pandemic such as being called “dirty” and accused of having the “China virus”. Participants also reported that other people physically distanced themselves farther than the amount recommended by public health authorities, which made them feel as if they were sick or as though they were the virus itself. It is also notable that some of the women described online encounters with such experiences; this corroborates other research during this period [10,29]. Studies have also reported that online victimization shot up alarmingly during the COVID-19 pandemic and negatively affected AAs’ mental health [30].

Regarding the themes related to self-reported changes in emotions during the pandemic, many respondents experienced emotional strain and social anxiety as they experienced overt and covert racial discrimination. These findings on emotional strain resonate with the results of previous studies that quantitatively demonstrated the negative impact of COVID-19-related racial discrimination on the mental health of AAs [10,14,15,18,19]. While these studies were limited to specific symptoms of mental health distress with established diagnostic criteria, such as depression and anxiety [14,18,31], our study expands on the current literature by adding more diverse types of emotional strains that AA women have experienced due to pandemic-related racial discrimination. In addition to common depressive symptoms such as feelings of hopelessness, worry, sadness, and frustration, this qualitative study captured other emotional strains such as fear, anger, and disappointment in American society.

For some respondents, these strains went beyond the emotional realm and brought about changes in behavior. Almost half of those who were categorized to *emotional strains* were also categorized to *social anxiety and limited social engagement* (*n* = 9). The reports in this theme correspond with the diagnostic criteria of social anxiety, which refers to feelings of discomfort in social settings and social encounters [32]. When social anxiety interferes with daily activities and impairs or prohibits normal functioning, individuals may be diagnosed with social anxiety disorder [31]. It is particularly noteworthy that AA women restricted their grocery shopping to Asian grocery stores, which indicates that AA communities may voluntarily segregate from other racial groups and mainly interact with other Asians to avoid discrimination. Previous research has shown that racially discriminatory comments from supervisors have lowered work engagement among AA women [33]. The current study broadens the current literature to show that the impact of COVID-19-related racial discrimination on AA women’s behavior applies to everyday life.

The themes that emerged concerning changes in racial identity were divergent and split, thus supporting both the AARID model and RIM [21,25]. The first two themes (i.e., *race hyperconsciousness* and *wakening to social–political positionality*) support the AARID model by illustrating how AA women’s racial identity was weakened due to prevalent racial discrimination and microaggressions. Indeed, the women became hyperconscious of how other people perceived them and their racial backgrounds, and some realized that they were perceived to be inferior and othered. The other two themes (i.e., *resiliency* and *empowerment*), in contrast, align with the RIM, illustrating how AA women developed a more resilient and empowered racial identity after experiencing COVID-19-related racial discrimination. These interesting findings on such nuanced changes in racial identity might not have been observed with a quantitative approach since the two directions of changes in racial identity could have offset each other and led to null findings [34]. Our findings indicate that the directions of change in racial identity among AA women could be two contrasting ways, supporting both the AARID model and RIM.

As previous studies have shown, racial identity may play a crucial role in determining levels of well-being among AAs in particularly hostile racial atmospheres [35,36]. The four themes of this study related to the changes in racial identity can be classified into two larger groups: (1) weakened racial identity (i.e., *race hyperconsciousness* and *wakening to social–political positionality*) and (2) strengthened racial identity (i.e., *resiliency* and *empowerment*). Both groups showed similar proportions of respondents experiencing *emotional strain*. Approximately 70% of the women from the weakened racial identity group and about 80% of those from the strengthened racial identity group cited such due to COVID-19-related racial discrimination. However, a significant difference was observed in the prevalence of *social anxiety and limited social engagement* experienced by these two groups. Approximately 50% of respondents with weakened social identity reported experiencing both *social anxiety and limited social engagement*, whereas only about 20% of respondents with strengthened racial identity reported the same. This observed pattern aligns with findings from previous studies among other racial minorities [37], which support the notion that racial identity mediates the relationship between perceived discrimination and mental health and well-being.

Despite novel findings, this study has several limitations. In studying the impact of COVID-19-related racial discrimination and microaggressions, other demographic determinants of the direction of racial identity change were not explored. In addition, the limited sample size and the cross-sectional nature of the data prevented further exploration of the intertangled relationships among racial discrimination, racial identity, and well-being. Considering the importance of racial identity in determining the mental health of racially minoritized individuals when they encounter racial discrimination, further research can identify determinants of the direction of this change using a larger sample. The recruited study participants mostly reside in the Boston metropolitan area, and this is not representative of Asian women in the region. The limitations in the sample may hinder the generalizability of the findings to AA women in Boston and AA women residing in other US regions.

Despite these limitations, the findings of this study provide implications for programs and policies to support AA women. For example, allyship is one of the impactful self-advocacies that can create a safe space for AA women individuals to share concerns and support each other, thereby building solidarity [38] as our participants desired. Advocacy for AA women can be implemented at a broader level, such as in educational settings, by providing culturally responsive curricula at institutions and training to educators and students. Such initiatives can promote understanding and respect among different racial groups, ultimately reducing anti-Asian violence against AAs. The legal requirement in public high schools to include ethnic studies, such as African American history and cultures of local Indigenous communities, in more than a dozen states (e.g., California, Illinois, New York, Florida) is one example highlighting the importance of educational advocacy [38,39]. Integrating these forms of advocacy into systematic approaches can contribute to dismantling the underlying structures of discrimination that perpetuate violence against AA women and promote healthy racial identity and well-being of AA women.

## 5. Conclusions

This study found that more than half of AA women experienced direct or indirect racial discrimination during the early COVID-19 lockdown. The hostile racial experiences impacted their well-being in largely negative ways. However, racial discrimination did not always negatively impact AA women’s racial identity.

Future research is needed to investigate several important questions on this topic. Specifically, how did racial identity impact mental health and physical health during the COVID-19 pandemic? What are the pathways to developing strong and healthy racial identities? Finally, we must build a society in which we no longer view race as a vulnerable characteristic but rather consider our unique talents, potential, and societal contributions as whole human beings.

## Figures and Tables

**Table 1 ijerph-20-06546-t001:** Demographic characteristics of study participants (*n* = 40).

	*n*	*%*
Discrimination experiences		
Have experienced racial discrimination after COVID-19 outbreak (yes)	14	35.00
Someone close has experienced racial discrimination after COVID-19 outbreak (yes)	19	47.50
Race and ethnicity		
Chinese	18	45.00
Korean	6	15.00
Vietnamese	2	5.00
Japanese	4	10.00
Filipino	1	2.50
Other Asian	2	5.00
Pacific Islander	1	2.50
Multiple Asian ethnicities	6	15.00
Nativity		
US-born	26	65.00
Foreign-born	12	30.00
Marital status		
Married	7	17.50
Living as married	1	2.50
Single/never married	31	80.00
Educational attainment		
High school or GED	2	5.00
Some college	7	17.50
College graduate	13	32.50
Postgraduate or professional schools	18	45.00
Household income in the previous year		
Less than $15,000	2	5.00
$15,000 to $25,000	3	7.50
$25,001 to $35,000	4	10.00
$35,001 to $50,000	5	12.50
$50,000 to $100,000	8	20.00
More than $100,000	16	40.00

**Table 2 ijerph-20-06546-t002:** Racial discrimination and microaggressions experienced by Asian American women.

Theme	*n*	Participants’ Quotes
*Verbal attack*	8	“Yes. I’ve been yelled at twice: once a stranger yelled mocking Chinese words as I left a store. another time, a person came up to my Asian friend and I and yelled for us to “go back to our country” and called us “dirty Asians” (22 years old, Japanese descent) “Two men standing outside of a restaurant on campus snickered and said “coronavirus” while I was walking to Student Health Services early March with a mask on” (22 years old, Chinese descent) “Racial epithets and slurs were thrown from people driving by while I am walking on the street” (31 years old, Korean descent
*Staring*	5	“Nothing very direct, maybe the occasional lingering stare” (22 years old, Chinese descent)
*Distancing or avoiding*	6	“Yes. I was at a grocery store in line to enter and a Caucasian middle aged man asked if he double his distance from me. He proceeded to check on me every time the line moved to ensure that I kept my distance” (28 years old, Chinese descent) “While getting in line at Trader Joe’s, an older white woman told me to “stay over there” because she was walking on the sidewalk” (42 years old, Chinese descent)
*Being* *followed*	2	“One time, I was on a walk outside with another Asian friend and a driver slowed down their car next to us, honked, and flipped us off. I don’t know whether it was racially motivated, but this is not something that ordinarily happens to me” (21 years old, Korean descent)
*Online* *discrimination*	2	“Online, have seen some harsh words. Most notable microaggression is “China virus” (23 years old, Chinese descent) “I am affected by what others post on social media, even in a “joking” manner. It hurts me very deeply and personally that people would perpetuate misconceptions or willfully ignore people’s feelings” (36 years old, Chinese descent)

**Table 3 ijerph-20-06546-t003:** Themes in changes in emotion, behavior, and well-being.

Theme	*N*	Participants’ Quotes
*Emotional strain*	22	“Generally feeling more hopeless and worried about the future” (21 years old, Chinese descent) “I feel a lot more sadness, anxiety, and fear than I did before” (21 years old, Korean descent) “I have been frustrated, angry, and disappointed at the outburst of racism and general discrimination I have seen in the midst of already incredibly trying times” (18 years old, Japanese descent)
*Social anxiety and* *limited social* *engagement*	11	“… scared to leave my house and interact with people” (28 years old, second-generation Chinese descent, and heterosexual) “I tense up whenever someone comes towards my direction. it’s harder to smile or say hello. while trying to avoid any potential episodes, i feel confined and limited” (32 years old, Korean descent) “I also only go to Asian grocery stores now because of my experience” (31 years old, Korean descent) “I was afraid to wear a mask in public to further call attention to myself as a potential target for attacks.“ (28 years old, Vietnamese descent)
*Concern for other Asians*	4	“I’m worried about myself and my Asian American friends. We look Asian, are we going to get harmed? Even if not physically, we’re worried a lot.” (23 years old, Chinese descent) “I feel sympathy for Asian Americans who have been the target of hate crimes, whether it was because of COVID-19 or not.” (23 years old, Hmong descent)
*Desire for solidarity*	2	“Overall, given me a desire to stand up against racism (toward all ethnicities) and do better to support those who are speaking out.” (23 years old, Filipino and Korean descent) “I want to continue speaking out against it instead of accepting the status quo.” (22 years old, Filipino descent)

**Table 4 ijerph-20-06546-t004:** Themes in changes in racial identity.

**Theme**	** *n* **	**Participants’ Quotes**
*Race hyperconsciousness*	13	“I became hyperaware of being Asian while out in public” (42 years old, Chinese descent)
*Awakening to* *social–political* *positionality*	11	***Inferior minority*** (*n* = 5) “It really emphasized for me the fragility of the model minority myth and exposed very early on that our perceived status as the “better people of color” is a lie” (19 years old, Chinese descent) “They simply see my Asian features. It’s really opened by eyes to the public’s perception of me and my racial background” (22 years old, Japanese descent) ***Otherness*** (*n* = 6) “Just like all other people of color, we are perceived as “other” in this country and, given how hard my family and I have tried to assimilate into American culture from the time I was born (in this country!!), that makes me very hurt” (19 years old, Chinese descent)
*Resiliency*	9	***Redirection to an Asian American consciousness*** “It for sure didn’t make me question my identity at all, I am a proud Asian American” (22 years old, Chinese descent) ***Incorporation*** “It hasn’t affected me. I (once lived in the Deep South, and) have an experience of being discriminated by both white and black people daily” (56 years old, Japanese and Pacific Islander descent)
*Empowerment*	4	“It has helped me realize that I should be a voice standing up against the racist comments” (23 years old, Filipino and Korean descent)

## Data Availability

The data presented in this study are available on reasonable request from the corresponding author. The data are not publicly available due to privacy restrictions on the IRB.

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
