# Peer review of "COVID-19-Related Racial Discrimination during Lockdown and Its Impact on Asian American Women"

_ijerph, 2023, doi:10.3390/ijerph20166546_

Round 1

Reviewer 1 Report

STRENGTHS

1. Excellent data, well explained and presented in the manuscript. 

MAJOR CONCERNS

1. An area being lacking in study is not strong enough a justification for the study (as stated in the abstract). Please avoid using this statement and provide a more academically sound argument for this manuscript in the abstract. Thank you. 

2. Please explain why thematic analysis was used as the data analysis tool as opposed to a coding process. Thank you.

3. In the abstract the author(s) state that "findings underscore the need for systematic forms of advocacy to combat anti-Asian racism and call for solidarity for AA women's wellbeing". However, in the manuscript the author(s) do not address advocacy. Rather, the authors have focused on implications for various stakeholders as well as ideas for future research. I would urge the author(s) to address this disparity between the content in the abstract and the manuscript. Specifically, please:

- Give suggestions for the type of advocacy for AA women in similar situations where Asians are targeted. 

- Address how these forms or types of advocacy would either offer resources or protections to AA during anti-Asian attacks.

- State clearly how these forms of advocacy may be systematic in nature to address anti-Asian racism targeted at Asian women.

Overall, a good paper and an interesting read but the authors need to address the issues raised above. Thank you.

MINOR CONCERNS:

Please watch out for minor spelling mistakes:

1. Page 3: "Breadth" and not "Breath". 

Please correct the minor concern mentioned in the comments to the author(s) above. Thank you.

Author Response

Dear Anonymous reviewer, 

Thank you so much for your thorough review. It was very helpful for us. Here are our responses to your concerns. 

MAJOR CONCERNS

  1. An area being lacking in study is not strong enough a justification for the study (as stated in the abstract). Please avoid using this statement and provide a more academically sound argument for this manuscript in the abstract. Thank you. 

Authors’ response: Thank you for your suggestion. Based on your comment, we modified the sentence to make it a more academically sound argument: “During the lockdown period of COVID-19 pandemic, Asian American (AA) women have experienced surge in anti-Asian hate crimes and racial discrimination, and majority of studies have quantitatively shown the negative impact of these incidents on Asian Americans’ well-being. Our research expands existing literature by qualitatively investigating types of COVID-19 racial discrimination during lockdown and its impacts on changes in emotions, behaviors, well-being, and racial identity development among AA women.” (pg.1, lines 13-18).

  1. Please explain why thematic analysis was used as the data analysis tool as opposed to a coding process. Thank you.

Authors’ response: We appreciate your critical comments on the study methodology. We chose thematic analysis as the data analysis tool since it is a higher-level analytical approach that enables researchers to synthesize and generate overarching themes, thereby providing a deeper understanding of the research question. We provided these descriptions in the method section (pg. 4, line 156-158).

  1. In the abstract the author(s) state that "findings underscore the need for systematic forms of advocacy to combat anti-Asian racism and call for solidarity for AA women's wellbeing". However, in the manuscript, the author(s) do not address advocacy. Rather, the authors have focused on implications for various stakeholders as well as ideas for future research. I would urge the author(s) to address this disparity between the content in the abstract and the manuscript. Specifically, please:

- Give suggestions for the type of advocacy for AA women in similar situations where Asians are targeted.

- Address how these forms or types of advocacy would either offer resources or protections to AA during anti-Asian attacks.

- State clearly how these forms of advocacy may be systematic in nature to address anti-Asian racism targeted at Asian

Authors’ response: We sincerely appreciate your critical points that can enhance our discussion section. While there is a lack of advocacy programs specific to AA women, we found some examples of advocacy that could apply to AA women. We also incorporated your suggestions in the paragraph as below:

Despite these limitations, the findings of this study provide implications for programs and policies to support AA women. For example, allyship is one of the impactful self advocacy that can create a safe space for AA women individuals to share concerns and support each other, thereby building solidarity [38] as our participants desired. Advocacy for AA women can be implemented at a broader level, such as in educational settings, by providing culturally responsive curricula at institutions and training to educators and students. Such initiatives can promote understanding and respect among different racial groups, ultimately reducing anti-Asian violence against AAs. The legal requirement in public high schools to include ethnic studies, such as African American history and cultures of local Indigenous communities, in more than a dozen states (e.g., California, Illinois, New York, Florida) is one example highlighting the importance of educational advocacy [38, 39]. Integrating these forms of advocacy into systematic approaches can contribute to dismantling the underlying structures of discrimination that perpetuate violence against AA women and promote healthy racial identity and wellbeing of AA women (pg. 11, line 387-399).

Reference:

[38] Wang, S.C.; Santos, B.M.C. At the intersection of the model minority myth and antiblackness: From Asian American Triangulation to Recommendations for Solidarity. J. Couns Psychol. 2023, 70 (4), 352–366.

[39] Kwon, S. Ethnic studies legislation: State scan. Available online: https://csaa.wested.org/wp-content/uploads/2021/03/ES-State-Scan-FINAL-v1.pdf (accessed on 23 July 2023).

MINOR CONCERNS:

Please watch out for minor spelling mistakes:

  1. Page 3: "Breadth" and not "Breath". 

Authors’ response: We made an edit accordingly (pg 2. Line 96). Thank you.

Reviewer 2 Report

Article is in great shape, clear, important, transparent.

A couple of minor questions - what was the age distribution in the respondent sample? This would be good to include in the reporting and discussion of the results as well.

Give a bit of a sense of how the Epi AWARE measurement/questions pivoted once the pandemic started; I'm assuming this went through IRB approval, so how did the research group change directions to include attention to this type of discrimination? If not, and the data was collected unrelated to COVID, explain in the article (maybe the first wave respondents offered a backdrop against which the wave two respondents were evaluated). Apologies if this is addressed already and I missed it, but I looked for this in a few places and don't find clarity; it seems the data is reported as if across both waves, but the questions seem different (p. 3).

Limitations seem a bit fuzzy, not as clear as expected, given the tone of the rest of the article.

Author Response

Dear anonymous reviewer, 

Thank you so much for your review! It is very helpful for us to improve this manuscript. Here are our responses to your review. 

Give a bit of a sense of how the Epi AWARE measurement/questions pivoted once the pandemic started; I'm assuming this went through IRB approval, so how did the research group change directions to include attention to this type of discrimination? If not, and the data was collected unrelated to COVID, explain in the article (maybe the first wave respondents offered a backdrop against which the wave two respondents were evaluated). Apologies if this is addressed already and I missed it, but I looked for this in a few places and don't find clarity; it seems the data is reported as if across both waves, but the questions seem different (p. 3).

Authors' response: Thank you so much for raising this question. We originally wrote a sentence but removed it because of internal concern that it may seem too redundant. Owing to your comments we added this paragraph to the methods section. Thank you very much for this opportunity to provide ample explanation of this project. 

Initially, Epi AWARE investigators planned to collect questionnaire data in two waves: at baseline (Wave 1 beginning December 2019) and follow-up at 6-months (originally designated as Wave 2 beginning June 2020). However, during the first wave of data collection, the Covid-19 outbreak occurred, followed by lockdown and a surge in anti-Asian hate crimes. In response to these environmental changes, the research team sought to understand the impact of this historic moment within our cohort by asking participants open-ended questions related to their contemporary experiences (newly designated as Wave 2). After the open-ended questions were collected, the team resumed its original plan of collecting follow-up data (now Wave 3) in September 2020. This research project was approved by the Institutional Review Board of the affiliated University Medical Center.

Limitations seem a bit fuzzy, not as clear as expected, given the tone of the rest of the article.

Authors' response: Thank you so much for your comments on our limitation. We added other limitations to our research in response to your comments. The final version of our limitation seems as follows. 

Despite novel findings, this study has several limitations. In studying the impact of COVID-19 related racial discrimination and microaggressions, other demographic determinants of the direction of racial identity change were not explored. In addition, the limited sample size and the cross-sectional nature of the data prevented further exploration of the intertangled relationships among racial discrimination, racial identity, and well-being. Considering the importance of racial identity in determining mental health of racially minoritized individuals when they encounter racial discrimination, further research can identify determinants of the direction of this change using a larger sample. The recruited study participants mostly reside in Boston metropolitan area and it is not representative of Asian women in the region. The limitation in the sample may hinder the generalizability of the findings to AA women in Boston and AA women residing in other U.S. regions.

Thank you very much! 

Best regards, 

Reviewer 3 Report

This is a well-written paper. It touches on a highly relevant topic using novel rich qualitative data. It checks about every box that an excellent research paper should check.   I do have some minor comments.    Alongside quotes, the authors reference subject's age, gender, generational status, and sexual identity. However, the authors do not discuss the significance of these intersectional identities or how they relate to the research findings. While I do believe including such information can provide a more comprehensive understanding of the participants' backgrounds and experiences, I also think it may be important for the authors to provide meaningful interpretation and analysis of these intersectional experiences. If the authors do not feel comfortable providing meaningful interpretation and analysis of these intersectional experiences, I suggest that the authors reconsider highlighting these intersectional identities as prominently.   While the authors briefly discuss the coding process, I would encourage much more detail here. This is imperative for promoting good scientific replicability.   I would also encourage the authors to better acknowledge the sample limitations and discuss how that may shape how readers interpret the results. The authors entirely neglect to acknowledge this sample is from only one metropolitan area and is not even necessarily representative of the population of interest in that metropolitan area.

Author Response

Dear Anonymous reviewer, 

Thank you so much for your thorough review! It was very helpful for us to improve the quality of this paper. Please find our responses to your comments below. 

Alongside quotes, the authors reference the subjects' age, gender, generational status, and sexual identity. However, the authors do not discuss the significance of these intersectional identities or how they relate to the research findings. While I do believe including such information can provide a more comprehensive understanding of the participants' backgrounds and experiences, I also think it may be important for the authors to provide meaningful interpretation and analysis of these intersectional experiences. If the authors do not feel comfortable providing meaningful interpretation and analysis of these intersectional experiences, I suggest that the authors reconsider highlighting these intersectional identities as prominently.  

-> Authors' response: We appreciate your critical comments. From all quotes, we removed information related to the sexual orientation of the participants. 

While the authors briefly discuss the coding process, I would encourage much more detail here. This is imperative for promoting good scientific replicability. 

-> Authors' response: Thank you for making emphasis on the scientific process of research. We add much more detail in the method section, particularly the analysis part. Below is what we included in the section. 

Free-text responses in Wave 2 data were analyzed using thematic analysis consistent with studies analyzing free-text survey responses [25, 26]. Thematic analysis is a high-level analytic approach that enables researchers to analyze open-ended data and identify overarching patterns that emerge from the data [25]. The data was divided into four different datasets according to the question to which responses were provided and then analyzed separately. Two researchers (ML and IYP) familiarized themselves with the data by reading all the responses repeatedly. Each researcher independently came up with initial codes that capture the most prevalent patterns emerging from the data. Through consensus meetings, two researchers (ML and IYP) created an initial codebook that includes the definitions and examples of each code and multiple codes given to individual responses as long as the codes appropriately cover the contents of responses. The codes of the two researchers were agreed upon 85% of the time, and inconsistency was resolved through further discussion in which the other authors (MP and HH) gave their third opinions. Codes were modified and finally generated into higher-order concepts (i.e., themes and major themes) based on their common properties through iterative discussions within the study team. (p. 4, lines 155-174)

 I would also encourage the authors to better acknowledge the sample limitations and discuss how that may shape how readers interpret the results. The authors entirely neglect to acknowledge this sample is from only one metropolitan area and is not even necessarily representative of the population of interest in that metropolitan area.

-> Authors' responses: We sincerely appreciate your critical perspectives toward our data. We agree with your opinion and added below as one of our limitations in the discussion section. 

The recruited study participants mostly reside in Boston metropolitan area and it is not necessarily representative of Asian women in the region. The limitation in the sample may hinder the generalizability of the findings to AA women in Boston and AA women residing in other U.S. regions. (p. 11, lines 380-384) 

Hope you find these changes appropriate. 

Best regards,